# Gamma Knife Radiosurgery for Cushing’s Disease: Evaluation of Biological Effective Dose from a Single-Center Experience

**DOI:** 10.3390/jcm12041288

**Published:** 2023-02-06

**Authors:** Yuan Gao, Mengqi Wang, Yang Wu, Hao Deng, Yangyang Xu, Yan Ren, Chun Wang, Wei Wang

**Affiliations:** 1Department of Neurosurgery, West China Hospital, Sichuan University, 37 Guoxue Alley, Chengdu 610000, China; 2Diabetic Foot Care Center, Department of Endocrinology and Metabolism, West China Hospital, Sichuan University, Chengdu 610000, China

**Keywords:** Cushing’s disease, gamma knife radiosurgery, safety, efficacy, biological effective dose

## Abstract

**Objective**: Gamma knife radiosurgery (GKRS) has served as an adjunctive treatment in Cushing’s disease (CD) for decades and has become a vital part of therapy in the management of CD. Biological effective dose (BED) is a radiobiological parameter with time correction, considering the cellular deoxyribonucleic acid repairment. We aimed to investigate the safety and efficacy of GKRS for CD and evaluate the association of BED and treatment outcome. **Methods**: A cohort study of 31 patients with CD received GKRS in West China Hospital between June 2010 and December 2021. Endocrine remission was defined as normalization of 24 h urinary free cortisol (UFC) or serum cortisol ≤ 50 nmol/L after a 1 mg dexamethasone suppression test. **Result**: The mean age was 38.6 years old, and females accounted for 77.4%. GKRS was the initial treatment for 21 patients (67.7%), and 32.3% of patients underwent GKRS after surgery due to residual disease and recurrence. The mean endocrine follow-up duration was 22 months. The median marginal dose was 28.0 Gy, and the median BED was 221.5 Gy_2.47_. Fourteen patients (45.1%) experienced control of hypercortisolism in the absence of pharmacological treatment, and the median duration to remission was 20.0 months. The cumulative rates of endocrine remission at 1, 2, and 3 years after GKRS were 18.9%, 55.3%, and 72.21%, respectively. The total complication rate was 25.8%, and the mean duration from GKRS to hypopituitary was 17.5 months. The new hypopituitary rate at 1, 2, and 3 years were 7.1%, 30.3%, and 48.4%, respectively. A high BED level (BED > 205 Gy_2.47_) was associated with better endocrine remission than a low BED level (BED ≤ 205 Gy_2.47_), while no significant differences were found between the BED level and hypopituitarism. **Conclusions:** GKRS was a second-line therapeutic option for CD with satisfactory safety and efficacy. BED should be considered during GKRS treatment planning, and optimization of BED is a potentially impactful avenue toward improving the efficacy of GKRS.

## 1. Introduction

Cushing’s disease (CD) is caused by prolonged supraphysiological level of adrenocorticotropic hormone (ACTH) from pituitary adenoma or hyperplasia, resulting in hypersecretion of cortisol from adrenal glands, which is related to hypertension, diabetes mellitus, centripetal obesity, osteoporosis, mood disorders, and even shortened life span [1]. The prevalence of CD is approximately 1.2–1.8 persons per million population per year, and CD is the main cause in approximately 70% of endogenous Cushing’s syndrome cases [2]. The diagnosis and management of CD is extremely complicated and requires multidisciplinary expertise, including neurosurgery, endocrinology, and cardiology. The goal of treatment is to normalize the hypercortisolemia, control tumor growth, relieve the damage to multiple systems and decrease the risk for unfavorable outcomes. Although surgical resection of ACTH-producing pituitary adenoma is the first-line therapy, 10–35% of patients with CD suffer either tumor recurrence or fail to achieve remission, particularly macroadenomas with suprasellar or cavernous sinus invasion [3,4,5,6]. In addition, the patients with severe coexisting disease could not tolerate the risk of anesthesia and surgery. Gamma knife radiosurgery (GKRS) has served as an adjunctive treatment in CD for more than 30 years and has gradually become a vital part of treatments in the management of CD.

The biological effective dose (BED) is a radiobiological parameter with time correction that considers cellular deoxyribonucleic acid repair during radiosurgery [7,8]. The overall treatment time influences the BED level directly. Generally, the longer the treatment time, the greater the opportunity for tissue repair; therefore, the BED decreases progressively [9]. Factors influencing the BED level included prescription dose, treatment dose rate, the number of isocenters, and the activity of the cobalt-60 sources [9]. Although numerous advancements have been made in GKRS systems and the accuracy and precision of GKRS have improved significantly, the relationship between BED and therapy outcome remains unstudied. The purpose of this study was to analyze the safety and efficacy of GKRS for CD and explore if BED can be a predictor of outcome after GKRS.

## 2. Materials and Methods

### 2.1. Patient Selection

Data were collected from 31 patients with CD who had undergone GKRS in West China Hospital of Sichuan University during June 2010 and December 2021. The inclusion criteria for this retrospective study included diagnosis of Cushing’s disease as determined by an endocrinologist and neurosurgeon, and follow-up duration ≥ 6 months after GKRS with sufficient data. The exclusion criteria for the study consisted of endocrine follow-up< 6 months, other types of pituitary adenomas, and deficient data regarding outcomes. Institutional review board approval was obtained for this study (2022/958).

### 2.2. Baseline and Follow-Up Data

The baseline data consisted of general conditions of the patients, tumor characteristics and GKRS parameters. General conditions of the patients included age, gender, coexisting disease, hormone levels of the pituitary-target organs prior to GKRS, and imaging data. Tumor characteristics included tumor volume and Knosp grade. The tumor volume was calculated as maximum height × maximum length × maximum width ×π/6. The GKRS parameters included marginal dose, maximum dose, isodose line, number of isocenters, and maximum point dose to the optic apparatus. Neuroimaging, biochemical tests, and complications were evaluated during the follow-up.

Endocrine remission was defined as normalization of 24 h urinary free cortisol (UFC) or serum cortisol ≤ 50 nmol/L after a 1 mg dexamethasone suppression test (1 mg DST). An increase in tumor volume ≥ 20% was defined as tumor progression, whereas a reduction in tumor volume ≥ 20% was defined as tumor regression, and the change between them was defined as stationary compared with pre-GKS volume.

### 2.3. Radiosurgical Technique

Twenty-six patients underwent the Leksell Gamma Knife (Elekta C) between June 2010 and December 2019, and five patients then received the Leksell Gamma Knife (Elekta ICON). First, a head frame was applied under local anesthesia. Second, thin slice (≤1 mm) MRI images were acquired with the frame in place. Then, the neurosurgeon and radiologist collaborated to establish the treatment plan. Generally, maximum doses to optic structures were kept to less than 9 Gy by optimized tumor treatment plans. Last, positioning the patient in the gamma knife, we delivered the plan after the head frame was rigidly affixed to the treatment table.

### 2.4. Biological Effective Dose Calculation

We calculated the BED via a simplified approach, which took into account the beam-on time, overall time, marginal dose, and isocenter, with an α/β ratio of Gy_2.47_ [9]. For the Elekta C system, the overall time included the beam-on time and additional between-shot time, which was calculated as the sum of 5 min per shot for n–1 shots. For the ICON system, the overall time was equal to the beam-on time. Then, we generated each estimated BED value using equation A9 reported in the study by Jones and Hopewell et al. [9].

### 2.5. Statistical Analysis

Statistical analyses were performed using SPSS version 23 (IBM Corp, Armonk, NY, USA). Descriptive statistics were performed for all available data. Kaplan-Meier and actuarial methods were used to perform time-dependent analyses for endocrine remission. A *p* value of <0.05 was defined as statistically significant.

## 3. Results

### 3.1. Patient Characteristics and GKRS Parameters

A total of 31 patients with CD received GKRS in this study (Table 1). The median age was 36 years old, and females accounted for 77.4%. Twenty-six patients (83.9%) suffered from coexisting disease. Hypertension was the most common disease (74.2%), and osteoporosis occurred in only five patients (16.1%).

GKRS was the initial therapy for 21 patients (67.7%), and 32.3% of patients underwent GKRS after surgery due to residual disease and recurrence. The median endocrine follow-up duration and radiological duration were 22 months (range: 6–126) and 18 months (range: 8–127), respectively. The median tumor volume was 608.0 mm^3^ before GKRS. The median marginal dose was 28.0 Gy (range: 16–33 Gy), and the median coverage percentage was 96.7%. The median maximum dose received by the optic apparatus and pituitary stalk was 6.8 Gy (range: 1.8–9.1) and 11.8 Gy (range: 3.6–25.1), respectively. The mean beam-on time was 1.5 h (range: 0.5–2.6), and the median treatment dose rate was 1.8 Gy/m (range: 1.3–2.9). The median BED was 221.5 Gy_2.47_ (65.3–316.7).

### 3.2. Endocrine and Imaging Outcomes

After a median follow-up duration of 22.0 months, the median levels of 24 UFC and cortisol after 1 mg DST decreased from 388.2 ug/24 h to 219.4 ug/24 h and 478.0 nmol/L to 45.1 nmol/L, as seen in Figure 1. Fourteen patients (45.1%) experienced control of hypercortisolism in the absence of pharmacological treatment, and the median duration to remission was 20.0 months (range: 6–64). By Kaplan-Meier analysis, the cumulative rates of endocrine remission at 1, 2, and 3 years after GKRS were 18.9%, 55.3%, and 72.2%, respectively, as seen in Figure 2(1). We analyzed the relationship between endocrine remission and the level of BED (BED > 205 Gy_2.47_ VS BED ≤ 205 Gy_2.47_). The actuarial rates of endocrine remission at 1 and 2 years after GKRS were 9.1% and 35.1%, respectively, in the low BED group (BED ≤ 205 Gy_2.47_); and 25.0% and 72.7%, respectively, in the high BED group (BED > 205 Gy_2.47_) (*p* = 0.041) as seen in Figure 2(2). A high BED level (BED ≤ 205 Gy_2.47_) was associated with better endocrine remission than a low BED level (BED > 205 Gy_2.47_). There were 21 patients with complete imaging data. After GKRS, tumor control was achieved in 20 patients (95.2%), and the median tumor volume decreased from 608 mm^3^ to 115 mm^3^. However, 1 patient (4.8%) suffered tumor progression (Table 2).

### 3.3. GKRS-Related Complications

The total complication rate was 25.8%. Of the eight patients (25.8%) who developed a new hypopituitary after GKRS, six patients had hypothyroidism, and two patients had growth hormone deficiency. The actuarial rates of the new hypopituitary rate at 1, 2, and 3 years were 7.1%, 30.3%, and 48.4%, respectively as seen in Figure 2(3). We also analyzed the relationship between new hypopituitarism and the level of BED (BED > 205 Gy_2.47_ VS BED ≤ 205 Gy_2.47_). The actuarial rates of the new hypopituitarism rate at 1, 2, and 5 years were 9.1%, 18.2%, and 45.5%, respectively, in the low BED group (BED ≤ 205 Gy_2.47_) and 4.1%, 35.2%, and 35.2%, respectively in the high BED group (BED > 205 Gy_2.47_) (*p* = 0.726) as seen in Figure 2(4). There was no significant difference between the BED level and hypopituitarism. The mean duration from GKRS to hypopituitary was 17.5 months (range: 5–59). No other neurological complications were encountered (Table 2).

## 4. Discussion

The main purpose of the treatment of patients with CD was to relieve hypercortisolism, resect the tumor mass, and preserve normal pituitary function [10]. Radiosurgery is considered an adjuvant therapy for the patients with CD and should be performed with accompanying medical therapy [11]. In addition, stereotactic radiosurgery can also be used as primary therapy in patients with high surgical risk or who refuse surgery [12]. We found that GKRS can relieve hypercortisolism and control the growth of tumors with satisfactory safety. Meanwhile, BED > 205 Gy_2.47_ was associated with higher endocrine remission rate without increasing the risk of hypopituitarism.

There was no consensus in the definition of biochemical remission for CD patients [12]. Several criteria to evaluate the efficacy of treatment have been made, including 24 UFC < 20 µg, serum ACTH < 5 pg/mL, and the normalization of midnight salivary cortisol [6,13,14]. Best biochemical remission for CD patients was a morning serum cortisol level less than 1 mg/dl on postoperative days 3, 4, or 5 [1]. However, the efficacy of GKRS could not be displayed immediately, and it took several months or years to reach remission for CD patients after GKRS. Therefore, the criterion of morning serum cortisol level was not suitable for CD patients to judge the outcome of GKRS. The normalization of 24 UFC has been widely applied in studies associated with GKRS [15,16,17]. Studies defining the remission criterion as the normalization of 24 UFC found that the endocrine remission rate ranged from 28.0 to 73.0% and that the median duration from GKRS to remission ranged from 10 to 36 months [15,16,17,18,19]. In the current study, in addition to the level of 24 UFC, serum cortisol ≤ 50 nmol/L after 1 mg-DST was applied to define remission, and our result corresponded with the literature [18,20].

Increasing evidence has shown that a high level of BED is a positive factor associated with endocrine remission. Anne et al. evaluated the BED in a series of 26 CD patients who underwent GKRS and defined the BED level as high BED (BED > 228 Gy_2.47_) and low BED (BED > 228 Gy_2.47_) [15]. They found that higher BED was associated with a higher overall biological remission for CD. In addition to prescription dose, BED should be considered during GKRS treatment planning. In this study, we found that high BED level (BED > 205 Gy_2.47_) was associated with better endocrine remission compared with a low BED level (BED ≤ 205 Gy_2.47_).

Hypopituitarism is a common radiation-associated adverse event for pituitary adenoma patients after GKRS. The hypopituitary rate ranged from 15.0 to 50.0% for CD after GKRS [16,17,18,21]. To avoid hypopituitarism after GKRS, the maximum dose, dose received by the pituitary stalk, tumor volume, and invasion of the cavernous sinus should be considered [22,23,24,25]. Compared with the patients with a history of surgical excision, the patients initially being treated with GKRS had a low risk of hypopituitarism [26]. For the patients with one or more prior resections, the chaotic anatomical structure can lead to the inaccurate delineation of the target and the inability to protect important structures such as the optic nerve, which was a possible reason for hypopituitarism. Whole-sellar stereotactic radiosurgery can increase the risk of hypopituitarism as well but is an alternative therapy when residual or recurrent tumors after surgery cannot be distinguished on neuroimaging [27]. The maximum dose to the gland and distal infundibulum played a vital role in the development of hypopituitarism. Marek et al. recommended keeping the mean dose to the pituitary gland under 15 Gy and the distal infundibulum dose under 17 Gy [28]. Graffeo et al. found that patients exposed to BED > 45 Gy_2.47_ were associated with a 14-fold increase in the risk of hypopituitarism compared with the patients exposed to BED ≤ 45 Gy_2.47_ [7]. They considered that the level of BED was more reliable for predicting hypopituitarism than the mean gland dose after SRS. However, in the current study, we defined a BED of 205 Gy_2.47_ as the cut-off point, and there was no significant difference between low BED and high BED. Although we applied the same methodology described by Jones and Hopewell, the results of BED exhibited discrepancies, as the minimum BED was 65.3 Gy_2.47_ in this study, which was significantly higher than that in the study by Graffeo et al. This suggests that the calculation method of BED needs further specification in detail.

Dose, dose rate, and any scheduled or unscheduled interruption of treatment determines the overall treatment duration, while the longer the overall treatment duration, the greater the chance of repair during exposure, which collectively alters the radiation biological effectiveness of the treatment. Voxel-by-voxel calculations can provide the real BED, rather than being based on whole-exposure volumes, with a daunting amount of calculation nevertheless. These estimation strategies must contain all kinds of assumptions which can lead to system errors. In practice, a balancing or trade-off, between accuracy and operability is often necessary [9]. A number of studies have adopted strategies by Jones and Hopewell; however, these methods have not been rigorously compared with other reported BED estimation methods for objective clinical validation.

Many studies have reported that GKRS was preferred for the CD patients accompanying severe coexisting. Gupta et al. reported that 21 patients with CD underwent initial GKRS, and the biochemical remission rate reached 81% at 5 years. In this study, 21 patients received primary GKRS, and one-third of them achieved remission. However, a subset of CD patients with severe symptoms needed to correct hypercortisolemia and provide rapid symptom palliation, and the drawback of delayed onset efficacy in GKRS was displayed. In addition to transsphenoidal surgery, bilateral adrenalectomy (BLA) was a definitive treatment to relieve to hypercortisolemia for patients with severe symptoms. Bunevicius et al. retrospectively included 50 patients with CD who underwent SRS before BLA [29]. The risk of developing to Nelson’s syndrome was reduced if patients with CD underwent SRS before BLA [29]. It appears that SRS should be considered in CD before proceeding to BLA [29]. Microadenomas are common in Cushing’s disease and it is beneficial in the treatment of GKRS.

## 5. Limitations

Although we found positive results in this study, several certain limitations should be mentioned. First, we drew a conclusion based on a limited sample size, which can influence the reliability of the results. Second, we found a significant difference between BED and treatment outcome, but we did not exclude the interference of confounding factors due to the limitation of sample size and the relationship between BED and outcome should be studied further. In addition, although the methodology of estimated BED reported by Jones and Hopewell was applied for several studies, the BED in this study was the approximate values for the entire treatment volume, instead of voxel calculations of the actual treatment BED.

## 6. Conclusions

GKRS is a second-line therapeutic option for CD with satisfactory safety and efficacy. BED should be considered during GKRS treatment planning, and optimization of BED is a potentially impactful avenue toward improving the efficacy of GKRS. BED > 205 Gy_2.47_ was associated with higher remission rate without increasing the risk of hypopituitarism. Further study should be performed to standardize the methodology of BED in detail and investigate the relationship between BED and the outcome of GKRS.

## Figures and Tables

**Figure 1 jcm-12-01288-f001:**
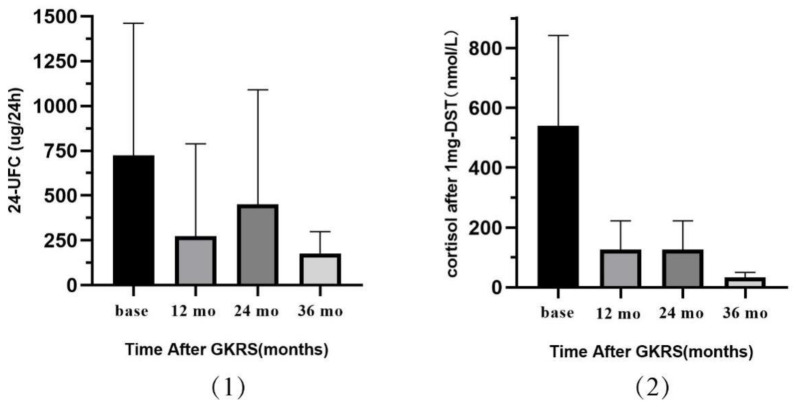
The trend of the 24 UFC (**1**) and cortisol after 1 mg DST (**2**) for patients with Cushing’s disease after receiving GKRS.

**Figure 2 jcm-12-01288-f002:**
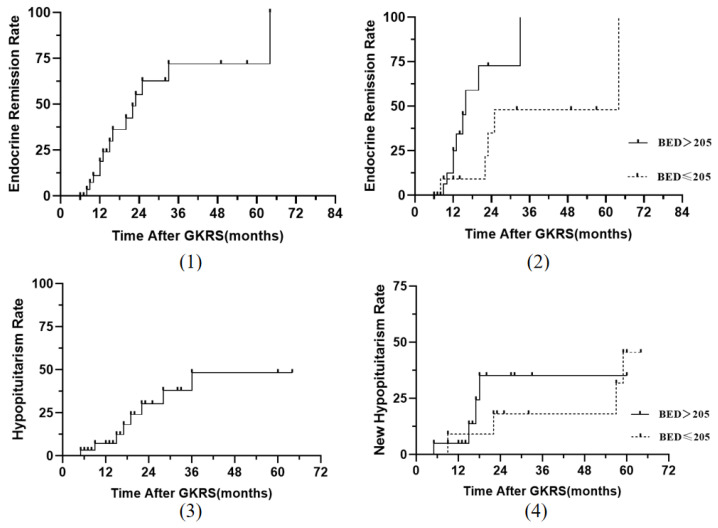
(**1**) Endocrine remission rate after GKRS for Cushing’s disease; (**2**) Kaplan-Meier analysis of endocrine remission after GKRS (*p* = 0.041, log-rank test); (**3**) new hypopituitarism rate after GKRS for Cushing’s disease; (**4**) Kaplan-Meier analysis of new hypopituitarism rate after GKRS (*p* = 0.726, log-rank test).

**Table 1 jcm-12-01288-t001:** Baseline characteristics of 31 patients with Cushing’s disease who underwent GKRS.

Variable	Value
Age at treatment (years), median (range)	36 (20–71)
Sex (M:F)	7:24
Coexisting disease	
Cardiac dysfunction	6/31 (19.4%)
Diabetes insipidus	18/31 (58.1%)
Hypertension	23/31 (74.2%)
Hyperlipidemia	8/31 (25.8%)
Bone rarefaction	5/31 (16.1%)
Radiosurgery indication	
Primary treatment	21/31 (67.7%)
Residual	7/31 (22.6%)
Recurrence	3/31 (9.7%)
No visible tumor pre-GKRS	2/31 (6.5%)
Cavernous sinus targeted	6/31 (19.4%)
Suprasellar component targeted	1/31 (3.2%)
Intrasellar	18/31 (58.1%)
Pre-GKRS tumor volume (median (IQR), mm^3^)	608.0 (462–978)
Pre-GKRS ACTH (median (IQR), ng/L)	61.63 (47.47–108.87)
Pre-GKRS cortisol after 1 mg DST (median (IQR), nmol/L)	478.0 (348.8–798.0)
Pre-GKRS 24 UFC (median (IQR), ug/24 h)	388.2 (257.9–998.9)
Endocrine follow-up duration (median (IQR), months)	22.0 (14–34)
Radiological follow-up duration (median (IQR), months)	18.0 (9–37)
Margin dose (median (IQR), Gy)	28.0 (26–31)
Maximum dose (median (IQR), Gy)	56.0 (51.8–62)
Isodose line (median (IQR), %)	50.0 (47–50)
Isocenters (median, IQR)	4 (3–6)
Coverage percentage (median (IQR), %)	0.967 (0.956–0.976)
Maximum optic apparatus point dose (median (IQR), Gy)	6.8 (5.0–8.0)
Maximum dose received by the pituitary stalk (median (IQR), Gy)	11.8 (8.8–17.8)
BED (median (IQR), Gy_2.47_)	221.5 (192.2–259.6)
TDR, (median (IQR), Gy/m)	1.8 (1.5–2.8)
Beam-on time (mean ± SD [range], hours)	1.5 ± 0.5 (0.5–2.6)

GKRS: gamma knife radiosurgery; ACTH: adrenocorticotropic hormone; 1 mg DST: 1 mg dexamethasone suppression test; 24 UFC:24 h urinary free cortisol; BED: biological effective dose; TDR: treatment dose rate.

**Table 2 jcm-12-01288-t002:** Treatment outcomes of 31 patients with Cushing’s disease who underwent GKRS.

Variable	Value
Endocrine remission rate	14/31 (45.2%)
Duration to remission (median (IQR) months)	20.0 (14.3–29.8)
Imaging regression rate	17/31 (81.0%)
Imaging progression rate	1/31 (4.8%)
Imaging stationary rate	3/31 (14.2%)
Last ACTH (median (IQR), ng/L)	47.0 (32.9–76.6)
Last cortisol after 1 mg DST (median (IQR), nmol/L)	45.1 (33.4–173.1)
Last 24 UFC (median (IQR), ug/24 h)	219.4 (145.0–595.3)
Total complications	8/31 (25.8%)
New visual disturbance	0/31 (0.0%)
Total new hypopituitary	8/31 (25.8%)
Hypothyroid	6/31 (19.4%)
Hypogonadism	0/31 (0.0%)
Hypocortisolemia	0/31 (0.0%)
Growth hormone deficiency	2/31 (6.5%)

ACTH: adrenocorticotropic hormone; 24 UFC: 24 h urinary free cortisol.

## Data Availability

All data generated or analyzed during this study are included in this published article.

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
