# Peer review of "Gamma Knife Radiosurgery for Cushing’s Disease: Evaluation of Biological Effective Dose from a Single-Center Experience"

_jcm, 2023, doi:10.3390/jcm12041288_

Round 1

Reviewer 1 Report

1. Comment to the tables 1 and 2: the name of the second column is "Total(%)", but not all data presented in this column match this name. Probably it is better to change the name of the column

2. It is not transcribed in the tables, what is IQR. If it is "Interquartile range", the data are expected to be presented as a range, but not a single figure

3. Could you please clarify why in the  majority of the patients included into the study, GKRS was the initial treatment. How many patients with CD in your center undergo  GKRS as the first line treatment  

Author Response

Response to Reviewer 1 Comments

Point 1. Comment to the tables 1 and 2: the name of the second column is "Total(%)", but not all data presented in this column match this name. Probably it is better to change the name of the column.

Response 1: Thank you very much for pointing out our mistakes in the table design. We have rearranged the tables 1 and 2 according to your suggestion to make it clearer as attachment showed.

Point 2. It is not transcribed in the tables, what is IQR. If it is "Interquartile range", the data are expected to be presented as a range, but not a single figure

Response 2:We have rearranged the tables 1 and 2 as response as attachment showed.

Point 3. Could you please clarify why in the majority of the patients included into the study, GKRS was the initial treatment. How many patients with CD in your center undergo GKRS as the first line treatment.

Response 3: GKRS is the second-line treatment of CD in our center, which is usually only suitable for patients who are not suitable or refuse surgery for various reasons. Most patients received surgery as initial treatment and achieved satisfactory results, while some number of patients failed to complete resection of the tumor or had tumor recurrence after surgery, so they received GKRS as second-line treatment. These 21 cases were CD patients who had received GKRS as initial treatment in our center in the past 11 years. They were not suitable for operation due to old age, underlying diseases, difficulty of operation, limited economic conditions and other reasons.

Reviewer 2 Report

The authors sholud note that the BED calculation is approximate and they should include a paragraph in the discussion about the great difficulty in calculating the BED.

In my opinion, they must emphasize that comparing BED in different series is extremely difficult

Author Response

We were pleased to accept the reviewer's helpful suggestions. We added the following paragraph to the conclusion to analyze the difficulties in calculating BED.

Dose, dose rate, and any scheduled or unscheduled interruption of treatment will determine the overall treatment duration, while the longer the overall treatment duration, the greater the chance of repair during exposure, which will collectively alter the radiation biological effectiveness of the treatment. Voxel-by-voxel calculations could provide the real BED, rather than based on whole-exposure volumes, with a daunting amount of calculation nevertheless. These estimation strategies must contain all kinds of assumptions which can lead to system errors. In practice a balancing, or trade-off, between accuracy and operability is often necessary. A number of studies had adopted strategies by Jones and Hopewell, however these methods have not been rigorously compared with other reported BED estimation methods for objective clinical validation.